# Phylogeny and molecular evolution of the first local monkeypox virus cluster in Guangdong Province, China

Jianhai Yu [1,7], Xin Zhang [2,3,7], Jiajun Liu [2,3,7], Linlin Xiang [1], Shen Huang [2,3], Xiaoting Xie [1], Ling Fang [2,3], Yifan Lin [1], Meng Zhang [2,3], Linqing Wang [1], Jianfeng He [2,3], Bao Zhang [1], Biao Di [4], Bo Peng [5], Jingtao Liang [6], Chenguang Shen [1] ✉, Wei Zhao [1] ✉ & Baisheng Li [2,3] ✉

The first local mpox outbreak in Guangdong Province, China occurred in June 2023. However, epidemiological data have failed to quickly identify the source and transmission of the outbreak. Here, phylogeny and molecular evolution of 10 monkeypox virus (MPXV) genome sequences from the Guangdong outbreak were characterized, revealing local silent transmissions that may have occurred in Guangdong whose mpox outbreaks suggested a molecular epidemiological correlation with Portugal and several regions of China during the same period. The lineage IIb C.1, which includes all 10 MPXV from Guangdong, shows consistent temporal continuity in both phylogenetic characteristics and unique molecular evolutionary mutation spectrum, reflected in the continuous increase of single nucleotide polymorphisms (SNPs) and shared mutations over time. Compared with the Japan MPXV, the Guangdong MPXV showed higher genomic nucleotide differences and separated 14 shared mutations from the B.1 lineage, comprising 6 non-synonymous mutations in genes linked to host regulation, virus infection, and virus life cycle. The unique mutation spectrum with temporal continuity in IIb C.1, related to apolipoprotein B mRNA-editing catalytic polypeptide-like 3, promotes rapid viral evolution and diversification. The findings contribute to understanding the ongoing mpox outbreak in China and offer insights for developing joint prevention and control strategies.

Since May 2022, the monkeypox virus (MPXV), previously endemic to East, Central, and West African countries, has widely spread and newly affected different countries and regions[1,2]. Unlike prior mpox outbreaks, the current outbreak exhibits epidemiological characteristics of broader and faster transmission, increased human-to-human sexual transmission, and accelerated viral mutation[3,4]. Although the number of reported cases in 2023 has markedly decreased compared to 2022, the spread of mpox is expanding, and the number of MPXV infections

[1]BSL-3 Laboratory (Guangdong), Guangdong Provincial Key Laboratory of Tropical Disease Research, School of Public Health, Southern Medical University, No. 1023, South Shatai Road, Baiyun District, Guangzhou, Guangdong Province 510515, China. [2]Institute of Microbiology, Center for Disease Control and Prevention of Guangdong Province, No. 160 Qunxian Road, Dashi Street, Panyu District, Guangzhou, Guangdong Province 511430, China. [3]Guangdong Provincial Key Laboratory of Pathogen Detection for Emerging Infectious Disease Response, No. 160 Qunxian Road, Dashi Street, Panyu District, Guangzhou, Guangdong Province 511430, China. [4]Department of Clinical Laboratory, Guangzhou Center for Disease Control and Prevention, No. 1 Qide Road, Baiyun District, Guangzhou, Guangdong 510440, China. [5]Shenzhen Center for Disease Control and Prevention, No. 8 Longyuan Road, Nanshan District, Shenzhen, Guangdong Province 518055, China. [6]Foshan Center for Disease Control and Prevention, No. 3 Yingyin Road, Chancheng District, Foshan, Guangdong Province 528010, China. [7]These authors contributed equally: Jianhai Yu, Xin Zhang, Jiajun Liu. ✉e-mail: a124965468@smu.edu.cn; zhaowei@smu.edu.cn; libsn@126.com

and deaths in newly affected countries and regions continues to increase. As of September 18, 2023, 115 countries and regions worldwide have reported ~90,439 cases and 157 deaths[5].

The first local mpox outbreak recorded in Guangdong Province was in June 2023. Within one month, the number of confirmed cases rapidly increased from 2 to 48, affecting several cities, including Guangzhou, Shenzhen, and Foshan. Local clustering outbreaks are often associated with sporadic importation infections due to travel to MPXV-endemic countries or personal contact with individuals infected with MPXV. However, available data for the epidemiological investigation to provide clues for controlling the spread of this outbreak were scarce. The findings suggested that all cases reported in Guangdong were local. Except for patient M23024, who traveled to Hong Kong on May 25, other patients had no overseas travel history from the Chinese Mainland; other than patients M23050 and M23108 who were confirmed as sexual partners, no connection between other patients was reported (Supplementary Data 1-Sheet 1). The tracing of transmission chains was incomplete, implying a certain level of uncertainty in promptly identifying the source of the epidemic in Guangdong and in gaining a thorough understanding of viral transmission based on the epidemiological background. It further underlines the necessity of implementing genome monitoring by public health organizations to characterize the epidemiology and evolution of MPXV[6,7].

Here, we report the whole genome sequences of ten local MPXV strains. Phylogenetic analysis was performed to determine the variant phenotype, phylogenetic position, and evolutionary trajectory, providing insights into the origin and transmission of this outbreak. Comparative genomic analysis was employed to investigate the genetic diversity, unique mutation spectrum, and microevolution-related recombination events to describe the molecular evolution of Guangdong MPXV.

## Results and discussion

### Phylogenetic analysis of lineage IIb C.1 containing all ten local MPXV from Guangdong

To rapidly characterize the phylogeny of the MPXVs involved in the outbreak in Guangdong, we constructed a phylogenetic tree by integrating 10 MPXV sequences from Guangdong into the global MPXV genetic diversity sequences derived from GISAID (Fig. 1a, Supplementary Data 1-Sheet 2 and Sheet 3). Focusing on lineage IIb C.1, MPXV isolated in 2022 and 2023 independently separated into two phylogenetic clusters with time continuity (Fig. 1b, c). The mpox in 2022 is characterized by widespread outbreaks in multiple countries and regions, mainly the United States, Ireland, the United Kingdom, and Italy. However, the mpox in 2023 spread to new countries or regions and cause local outbreaks, such as South Korea, Japan, China, and Portugal (Fig. 1d). Although the latest World Health Organization report states that the number of mpox reported cases in 2023 has markedly decreased compared to in 2022[5], vigilance, especially in the countries and regions where the first mpox cases are reported, is necessary as silent local transmission may lead to a rapid increase of MPXV infection cases[8].

All ten local MPXVs belonged to the IIb C.1 lineage, forming an individual cluster with the MPXVs from outbreaks in Portugal and in Zhejiang and Yunnan Provinces of China during the same period (Fig. 1e). This cluster differed from those of strains introduced into China earlier (Chongqing: B.1[9], Hong Kong: B.1.7[10], and Taiwan: B.1.5[11]) and had the closest phylogenetic relationship with TMIPH0076 strain (GISAID ID: EPI_ISL_17692269) from Japan collected on April 28, 2023 (Fig. 1e). Notably, in June 2023, China reported two local mpox cases, for the first time, and the phylogenetic analysis of these two MPXVs was linked to imported cases reported in Beijing, also indicating an association with the TMIPH0076 strain from Japan[12].

Notably, the collection date of the two cases from Beijing was the earliest in the Chinese Mainland (case 1: May 31, 2023; case 2: June 2,

2023). However, in our study, all ten patients reported no travel history to Beijing, and the MPXV of these two Beijing cases was not included in the phylogenetic analysis because they had not been uploaded to the public database. Therefore, it is not yet possible to determine the true correlation between local mpox outbreaks in Guangdong and Beijing. However, in this study, although both the MPXVs in Zhejiang and Yunnan were found to share a highly consistent phylogenetic position with that in Guangdong, the HZCDC-001 strain (GISAID ID: EPI_ISL_17809521 [https://www.epicov.org/epi3/frontend#290a7a]) from Zhejiang Province was the more possible source of the outbreak in Guangdong (Fig. 1e). These findings suggest potential epidemiological connections between MPXVs in different regions of China during the same period and association with the mpox in Japan in 2023. These newly affected regions in the Chinese Mainland possibly undergo silent local transmission. However, whether this local outbreak in Guangdong resulted from the importation event that occurred in Beijing or multiple transmission chains caused by different importation events from Japan, remains uncertain. Furthermore, between June 2, 2023, and August 30, 2023, mpox spread rapidly in the Chinese Mainland, with reports of 1098 confirmed cases across 25 provinces and regions[13–15]. Since MPXV sequences from many regions have not been uploaded to the public database, obtaining MPXV genomes from respective regional outbreaks for phylogenetic characterization is necessary to determine the spreading network of the ongoing mpox outbreak in the Chinese Mainland.

### Microevolutionary characteristics of 10 local MPXVs from Guangdong

Single nucleotide polymorphisms (SNPs) are considered key drivers in the rapid evolution and adaptive changes of poxviruses[16]. Compared to the reference strain of lineage B.1 (GenBank: ON563414.3), the 10 MPXV sequences of Guangdong diverged with a mean of 21 SNPs and displayed 39 distinct SNPs (comprising 23 non-synonymous and 16 synonymous mutations) (Fig. 2a). All 39 SNPs were divided into 14 shared mutations, 13 partially shared mutations, and 12 private mutations based on the mutation frequency shared in the 10 MPXV sequences of Guangdong (Fig. 2b). Epidemiological investigations confirmed that the MPXV genomes of patients M23050 and M23108, identified as sexual partners, were completely identical, whereas the MPXV from the other eight self-reported unrelated patients displayed several private mutations (Fig. 2a, Supplementary Data 1-Sheet 4). These private mutations specific to individual cases potentially occurred after several transmissions, suggesting the possibility of silent transmission of mpox within Guangdong.

Further analysis of the mutation spectrum of the 39 SNPs revealed that 24 were apolipoprotein B mRNA-editing catalytic polypeptide-like 3 (APOBEC3-like) mutations, i.e., TC>TT and GA>AA nucleotide substitutions, which play a role in virus genome editing and facilitate adaptive mutations in MPXV during rapid transmission (Fig. 2c)[6,17]. Among the 24 APOBEC3-like mutations, 11 non-synonymous mutations were located in genes encoding proteins with different functions, such as host immune regulation (OPG003, OPG005, OPG025, OPG036, and OPG176), surface membrane proteins (OPG074), and viral transcriptional regulation (OPG124, OPG178, and OPG180) (Fig. 2d). Notably, all 14 shared mutations among 10 MPXVs in our study were APOBEC3-like mutations, while 15 non-APOBEC3-like mutations out of 39 SNPs only appeared in partially shared and private mutations (Fig. 2e). This phenomenon was also observed in the investigation of the B.1 lineage, which showed an outbreak in 2022 but in lower proportions (5/46[17] and 5/22[18]). Therefore, compared to the B.1 lineage implicated in the mpox outbreak of early 2022, these unique features of shared, partially shared, and private mutations possibly facilitated the rapid evolution and diversification of the current MPXV lineage[19]. Moreover, the C151626T mutation of OPG178 (A49R) in the MPXV strain derived from patient M23008 resulted in a premature stop codon at the 40th amino

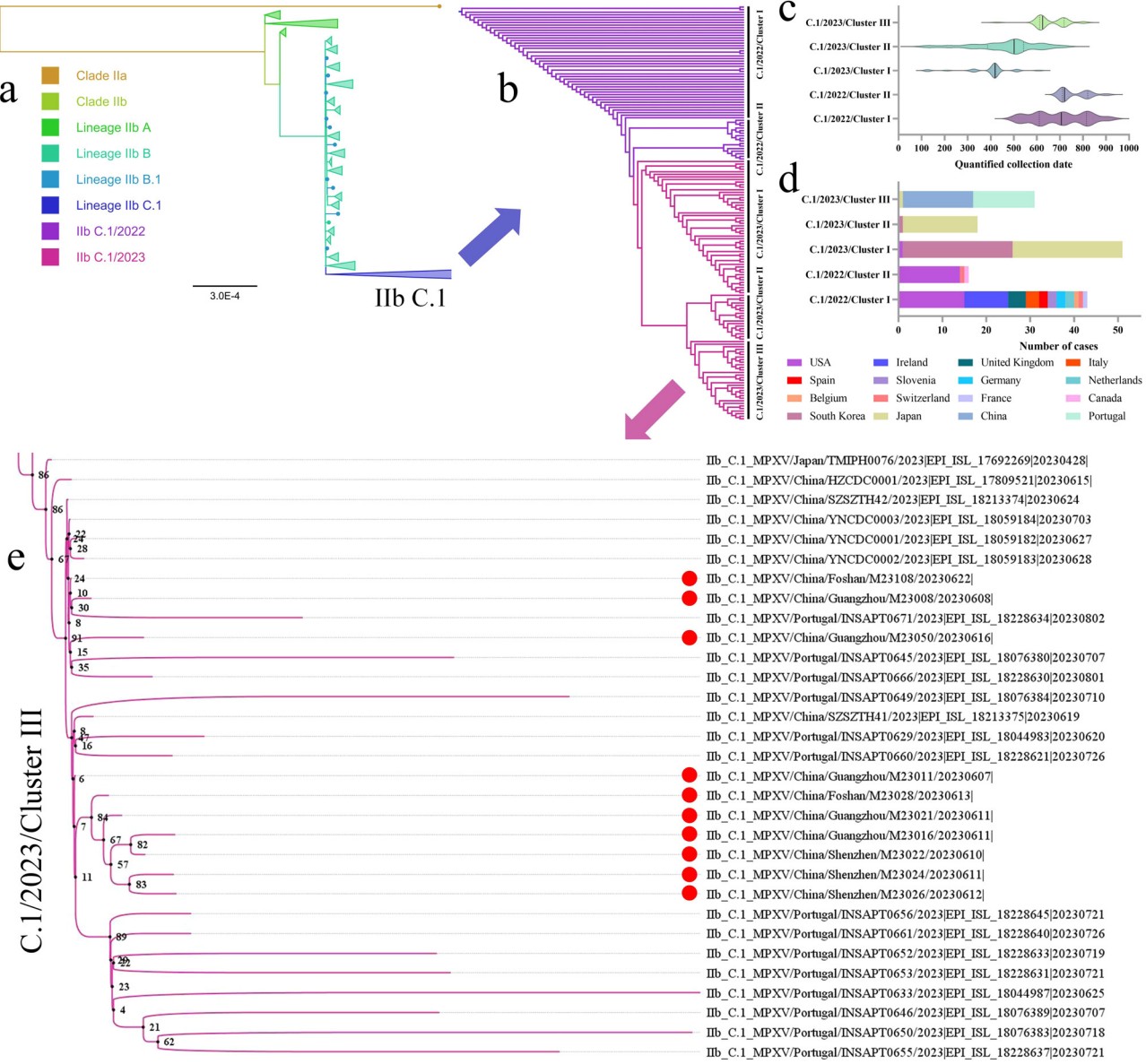

**Fig. 1 | Phylogenetic analysis of the MPXV genome sequences associated with the local Guangdong outbreak in 2023, as of September 12, 2023. a** A global phylogenetic tree that integrated the 10 MPXV genome sequences outbreak in Guangdong. The clade and lineage information comes from the GISAID database, and all MPXV in Guangdong belong to the IIb C.1 lineage. **b** MPXV clusters with time continuity divided from the IIb C.1 lineage. **c** Time continuity characteristics of MPXV clusters in the IIb C.1 lineage. The collection date of MPXV is quantified and displayed in a four-digit format consisting of months and days. The black solid and dashed lines represent median and quartile, respectively. **d** The source country distribution of MPXV clusters in the IIb C.1 lineage. **e** The phylogenetic characteristics of C.1/2023/Cluster III including all ten MPXV from Guangdong. The black small circle represents the node of the phylogenetic tree, the number represents the reliability (%) of the node, and the red circle represents the MPXV from Guangdong in this study.

acid of the encoded protein A49R (Fig. 2a, d). Additional studies are required to verify whether the premature termination of A49R affects viral replication and transcription.

### Mutation spectrum with time continuity and recombination analysis in the IIb C.1 lineage of MPXV

We further explored the molecular evolution characteristics of lineage IIb C.1. Based on the phylogenetic positions with time continuity, the MPXVs of C.1 were first divided into two clusters: C.1/2022 and C.1/2023. Next, C.1/2022 was further divided into C.1/2022/cluster I and C.1/2022/cluster II, representing early widespread outbreaks in multiple countries and later localized outbreaks mainly in the USA, while C.1/2023 was divided into C.1/2023/cluster I, C.1/2023/cluster II, and C.1/2023/cluster III, respectively, representing early co-outbreaks between South Korea and Japan. Subsequently, localized outbreaks in Japan and the latest outbreaks in China and Portugal have been reported (Fig. 1b–d). Over time, a continuously increasing trend in the number of nucleotide substitutions in an MPXV was observed, described as SNPs separated from MPXV in 2023 being significantly higher than that in 2022. The SNPs of latter cluster is also significantly higher than that of the previous one in the same year, while no difference exists in the latest MPXVs between China and Portugal during the same outbreak period (Fig. 3a).

When focusing on shared nucleotide mutations separated from the reference genome of the B.1 lineage, the above MPXV clusters, divided by phylogenetic positions, showed a unique mutation spectrum. All these mutations were found in APOBEC3-like and exhibited continuous molecular evolution: one shared mutations with the C.1/

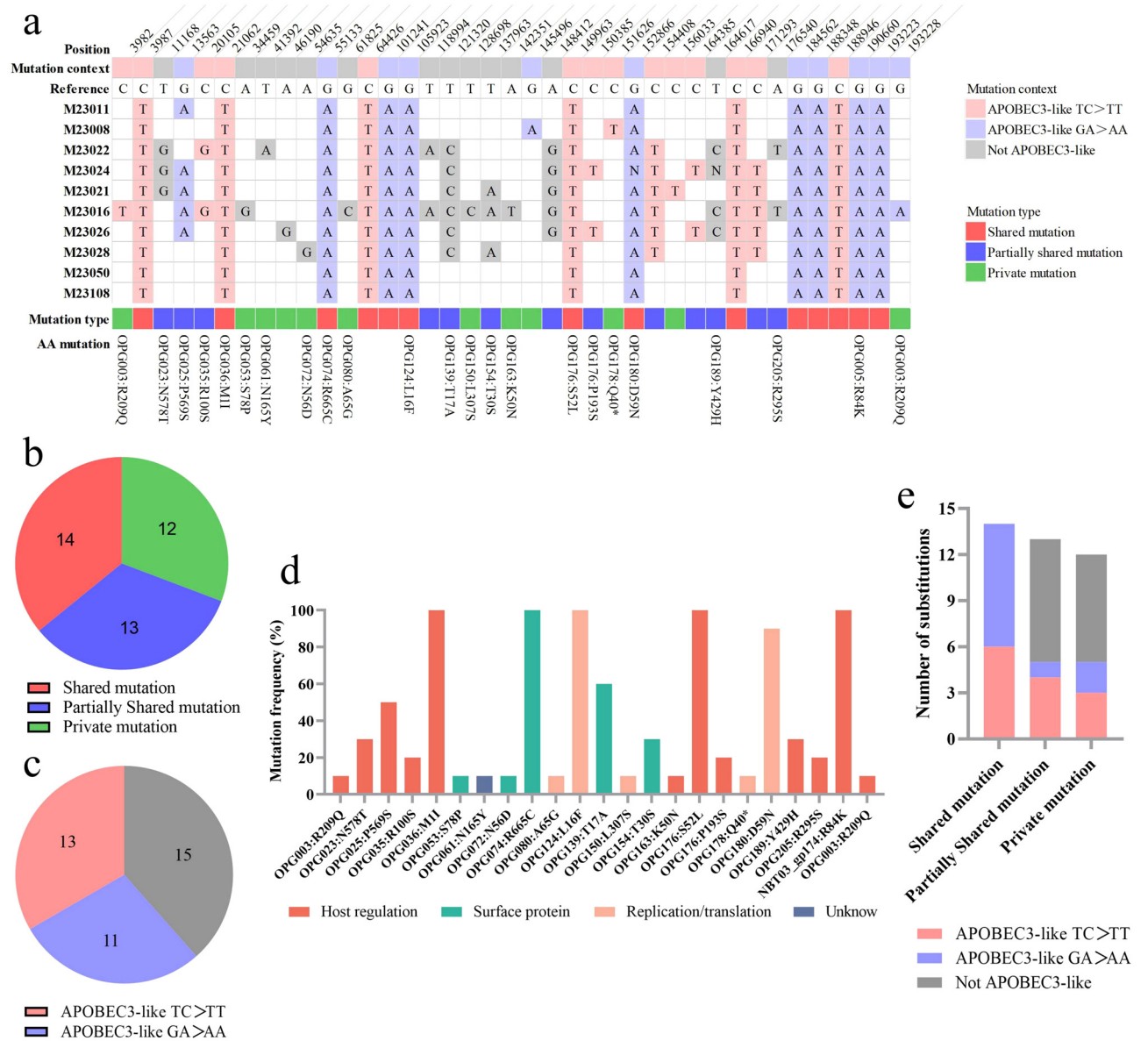

**Fig. 2 | Molecular evolution characteristics of ten local MPXV from Guangdong 2023. a** A total of 39 distinct SNPs were separated by the Guangdong MPXV from the reference genome sequence of IIb B.1 (GenBank: ON563414.3) in NC_063383.1 coordinates. Mutation context means Nucleotide substitution type and the classification of mutation types is based on the mutation frequency shared in the 10 MPXV sequences of Guangdong. **b** Distribution of mutation types in the 39 SNPs. **c** The nucleotide substitution distribution of the 39 SNPs. **d** Mutation frequency and protein annotation of 23 non-synonymous mutations in 39 SNPs. 'Unknown' means the GenBank function of this protein was annotated as unknown in the reference MPXV (GenBank: NC_063383.1). **e** The nucleotide substitution distribution of different mutation types. All 14 shared mutations among the Guangdong MPXVs were APOBEC3-like mutations.

2022/cluster I; three with the C.1/2022/cluster II; seven with the C.1/2023/cluster I; eight with the C.1/2023/cluster II; and 14 with the C.1/2023/cluster III (Fig. 3b, Supplementary Data 1-Sheet 5). Notably, unlike Guangdong, Yunnan, and Portugal MPXV, which also belong to C.1/2023/cluster III wherein the same 14 shared mutations were observed, only 12 were separated from Zhejiang MPXV, and its phylogenetic location was closer to the Japanese TMIPH0076 strain (Fig. 1e). This finding suggests that the outbreak in Zhejiang may have occurred earlier than in Guangdong and Yunnan during the transmission of mpox in China; however, due to the lack of available information on Beijing MPXV, a complete transmission network could not be determined. Furthermore, the molecular evolution and phylogenetic characteristics of the MPXV isolated from Zhejiang, Yunnan, and Guangdong demonstrated remarkable consistency, offering new evidence to support the hypothesis of silent local transmission and

independent evolution in the mpox from China. Compared to the Japan MPXV of C.1/2023/cluster II, the Guangdong MPXV of C.1/2023/cluster III added seven shared mutations among all infected individuals. This group of mutations comprised three non-synonymous mutations: OPG036:M1I, OPG176:S52L, and OPG180:D59N. These genes are involved in the transcription and replication of MPXV, as well as in the host's immune regulation, and they constitute the distinct adaptive mutation spectrum of the MPXV in Guangdong (Fig. 3b, Supplementary Data 1-Sheet 5).

Yeh et al.[20]. were the first to observe recombination events in MPXV strains from the outbreak in 2022 using the tandem repeat and linkage disequilibrium model. A similar recombination phenomenon was also observed in clade IIb B and IIb C using the Recombination Detection Program 4 (RDP4) recombination analysis software and the recombination event was found in almost all IIb B.1 and its derived

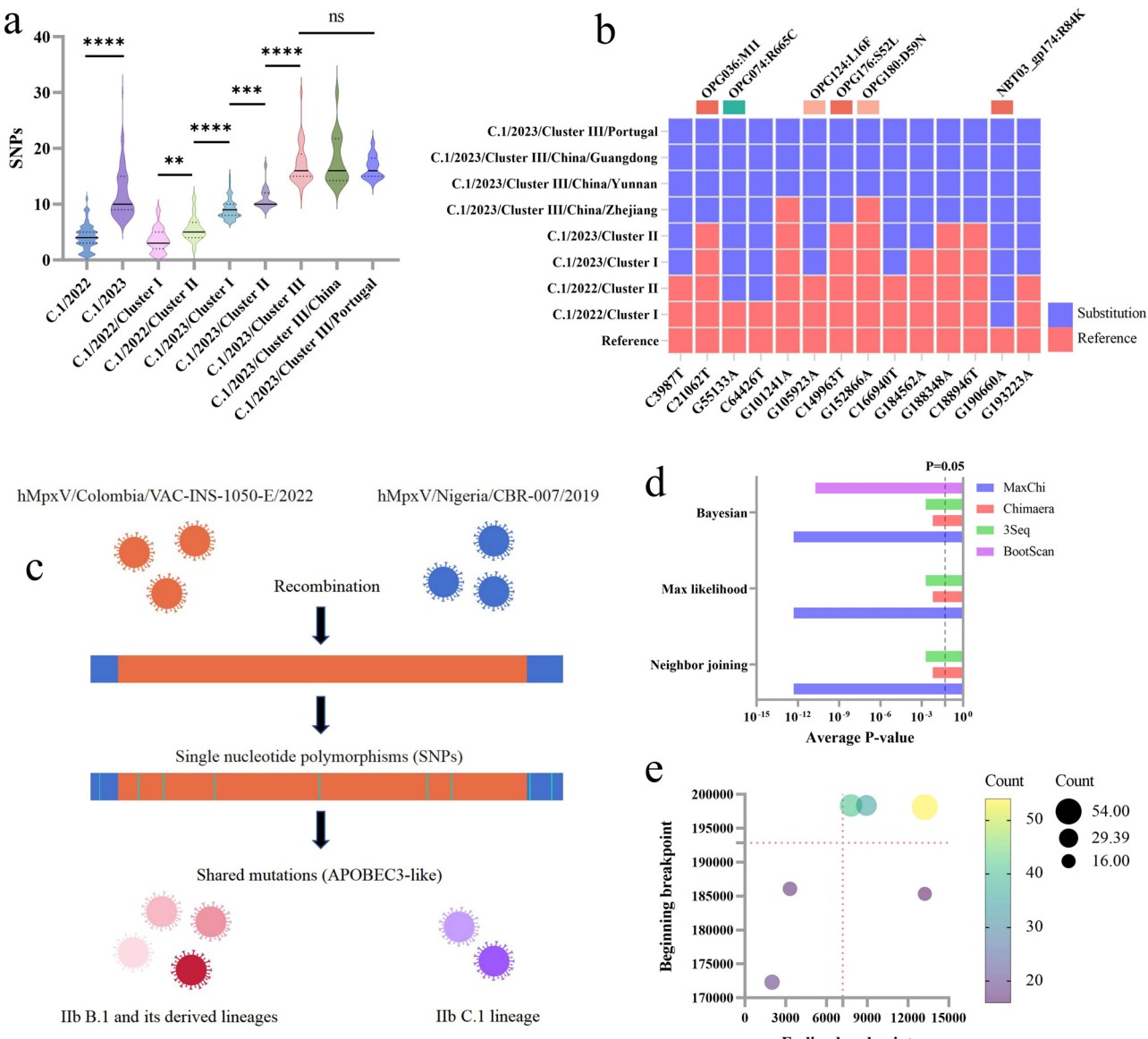

**Fig. 3 | Molecular evolution characteristics of lineage IIb C.1. a** The SNPs separated from the reference genome sequence of IIb B.1 (GenBank: ON563414.3) in the MPXV clusters of the IIb C.1 lineage. The black solid and dashed lines represent median and quartile, respectively. Shapiro-Wilk normality tests were used to verify whether SNPs follow a normal distribution, and two-sided independent sample T-test and One way ANOVA with Tukey's multiple comparisons test were used for comparison between two groups and multiple groups, respectively. ns: $P < 0.05$; **$P < 0.01$; ***$P < 0.001$; ****$P < 0.0001$. **b** Molecular evolution of unique mutation spectrum from the MPXV clusters of lineage IIb C.1. The sequence of ON563414.3 in NC_063383.1 coordinates served as reference, with a color change from red to blue indicating a mutation. All these mutations were found to be APOBEC3-like

mutations and exhibited continuous molecular evolution. **c** A recombination event observed in almost all of the IIb B.1 and its derived MPXV lineages, as well as C.1 lineage. **d** The reliability of the recombination event was demonstrated by constructing three evolutionary trees using seven recombination analysis methods. The recombination analysis method without displaying $P$ values indicates that it is not statistically significant ($P > 0.05$), while $P < 0.05$ indicates support for the occurrence of the recombination event. **e** Distribution characteristics of six pairs of beginning and ending breakpoints with a frequency exceeding 5% ($n > 12$). The red dashed lines on the $X$ and $Y$ axes were used to mark the repetitive regions of variable length, which were 0-7192 and 192856-200578 in alignment with Supplementary Data 3.

MPXV lineages, as well as IIb C.1 (Fig. 3c, d, Supplementary Data 1-Sheet 3). Notably, the regions where recombination events occurred were most located at the inverted terminal repeat of the MPXV genome and most breakpoint positions (beginning or ending breakpoints) were within the repetitive region of variable length (Fig. 3e, Supplementary Data 1-Sheet 6). This region and breakpoint are considered an important target for the recombination of poxvirus due to its high mutation rate poxvirus reported previously[21,22]. The sequence used to infer the major parent of this recombination was hMpxV/Colombia/VAC-INS-1050-E/2022 of the IIb B.1.16 lineage (GISAID ID: EPI_ISL_15802710

[https://www.epicov.org/epi3/frontend#290a7a]), and the minor parent was the strain hMpxV/Nigeria/CBR-007/2019 of the IIb A (GISAID ID: EPI_ISL_15370066 [https://www.epicov.org/epi3/frontend#290a7a]) (Fig. 3c). This can be inferred as the Colombian MPXV which was detected in early mpox in 2022 may have undergone a recombination event with the MPXV in Nigeria in 2019, causing a widespread global outbreak, thus retaining this event on almost all IIb B.1 and its derived MPXV lineages, as well as IIb C.1. This finding suggests that early recombination events in the IIb cluster might have accelerated the evolution and spread of the virus, which may be associated with the

epidemiological changes in IIb B.1 and its derived lineages, and underlies the reason for increasing cases of MPXV in newly affected countries.

In rapidly evolving poxvirus, adaptive changes in the genome may be driven by two mechanisms, namely recombination and SNPs[23]. Recombination is more conducive to generating new genetic phenotypes and greatly enhances the potential for population transmission and pathogenicity[20,24]. The latest research suggests that the host APOBEC3-like deaminase may contribute to the accelerating evolution of SNPs, as the shared mutations of current MPXV new mutants all contain this nucleotide substitution mechanism[17]. Based on recombination and SNP analyses (Fig. 3a–c), we inferred that the MPXV outbreak in 2022 may have originated from a single recombination event, and accelerated the mutation of MPXV under the APOBEC3-like nucleotide replacement mechanism, thus evolving into different lineages and subpopulations (Fig. 3c). However, more evidence is needed to prove the true occurrence of recombination, clarify how the editing of host genes drives viral mutations, and whether other mechanisms drive viral mutations, as the proportion of non-APOBEC3-like mutations in SNPs of MPXV found currently in the outbreak in China and Portugal has significantly increased.

In summary, this is the first reported phylogeny and molecular evolution of local MPXV cluster in China. Our findings suggest that the mpox outbreak in Guangdong Province was not related to previous importation events in China but was associated with the outbreak in Japan in 2023, potentially caused by single or multiple importation events leading to the local cluster outbreak. The local MPXV in Guangdong may have been silently transmitted in the population, separating more SNPs from the B.1 lineage and presenting more complex genomic polymorphisms and a unique adaptive mutation spectrum compared to the 2023 outbreak in Japan. The identified recombination and mutations may contribute to clarifying the process in the generation and accelerated evolution of the IIb C.1 lineage that causes current mpox and offer insights into the epidemiological and molecular evolutionary features of the local cluster outbreak of mpox in Guangdong, thus contributing to guiding prevention and control strategies in the current mpox outbreak. Furthermore, concurrent mpox outbreaks in Guangdong, Yunnan, Zhejiang, and Beijing might imply potential epidemiological correlations, suggesting a more severe situation with silent transmission in China Mainland. This situation calls for more comprehensive epidemiological investigations and rigorous MPXV genomic monitoring in collaboration with public health departments across other regions to promptly mitigate the mpox outbreak.

## Methods

### Ethics statement

This study was approved by the Ethics Committee of Guangdong Provincial Center for Disease Control and Prevention (Guangdong CDC) and complies with all relevant ethical regulations. The epidemiological investigations, sample collection, and viral isolation and cultivation were conducted by Guangdong CDC with the informed consent of the patient, following the Guidelines for the diagnosis and treatment of mpox (2022 edition) issued by the National Health Commission of China. The data analysis process is anonymous. Any information involving the patient's privacy, such as name, ID card number, telephone number, age, etc., was deleted by Guangdong CDC before analysis and was made strictly confidential to the data analyst of this study. The publication of epidemiological personal data in Supplementary Data 1-Sheet 1 complied with relevant ethical regulations and has been approved by the Guangdong CDC.

### Genome sequence of MPXV in Guangdong Province

Ten local MPXV whole genome sequences from outbreaks in Guangdong were included in this study. All MPXV whole genomes were sequenced using second-generation sequencing technology on an Illumina Miniseq instrument. The MPXV DNA was extracted with the CqEx-DNA/RNA kit (Tianlong, China, Catalog Number: 22102110T333), amplified to 2500 kb fragment by PCR using the Monkeypox Whole Genome Sequencing kit (Cyanines, China, Catalog Number: RBK-MPXV-2500), and sheared to ~500 bp fragments using the Nextera XT DNA Library Prep kit (Illumina, USA, Catalog Number: 15032354). Assembly was performed using the IPH-nano sequencing analysis software with the reference genome of strain, MPXV-M5312_HM12_Rivers (lineage IIb A, GenBank Accession No. NC_063383.1 [https://www.ncbi.nlm.nih.gov/nuccore/NC_063383.1]).

Finally, detailed data on genome assembly, including coverage, the total number of reads, read length, and mean depth of coverage, as well as the epidemiological background of the patients, were compiled, as shown in Supplementary Data 1-Sheet 1.

### Phylogenetic analysis

The local MPXVs of Guangdong were preliminarily confirmed to belong to the IIb C.1 lineage using NextClade v2.14.1 (https://clades.nextstrain.org/). Subsequently, a global MPXV genetic diversity sequence dataset was created based on the GISAID public database (https://www.epicov.org/epi3/frontend#290a7a). Using the setting conditions "complete," "high coverage," "low coverage exclude," and "coll date complete," as of September 12, 2023, 1703 whole MPXV genome sequences were extracted from the GISAID database (Supplementary Data 1-Sheet 2). For the clade and lineage information of the 1703 MPXV sequences, a random number method was used to select 10 MPXVs from each clade and lineage into the dataset; if fewer than 10 were available, all were included. All 151 MPXVs from the IIb C.1 lineage, as well as the reference strains MPXV-M5312_HM12_Rivers (lineage IIb A, GenBank Accession No. NC_063383.1) and MPXV_USA_2022_MA001 (lineage IIb B.1, GenBank Accession No. ON563414.3) from the NCBI virus database, were also included. Finally, the 10 MPXV sequences from Guangdong were integrated into the dataset (Supplementary Data 1-Sheet 3). Rapid sequence alignment was achieved using the "mafft --add module" of MAFFT v7 (https://mafft.cbrc.jp/alignment/server/add_sarscov2.html?mar15). Non-coding and terminal repeat regions were masked using MEGA 11 (https://www.megasoftware.net/). The final aligned dataset provided as Supplementary Data 2 was then utilized to construct a maximum likelihood phylogenetic tree using the best-fit K3Pu+F+I substitution model with IQTREE v2.2.2.6 (http://www.iqtree.org/), and the node support rate (%) was calculated by ultrafast bootstrap approximation method with 1000 repetitions. The phylogenetic tree was visualized using FigTree v1.4.4 (http://tree.bio.ed.ac.uk/software/figtree/).

### Molecular evolution analysis

Mutation sites were extracted and visualized from the sequence alignments of the lineage IIb C.1 using NextClade v2.14.1 (https://clades.nextstrain.org/) with strain MPXV_USA_2022_MA001 (reference of lineage B.1) in NC_063383.1 coordinates as the reference strain. The relevant MPXV sequence information is provided in Supplementary Data 1-Sheet 3. Based on the dataset used to construct phylogenetic trees, NextClade was used to select MPXV sequences with good quality control and over 99% coverage for recombination analysis (Supplementary Data 1-Sheet 3). A total of 259 MPXV sequences were subjected to sequence alignment through MAFFT v7 and their non-coding regions were masked through MEGA 11. Finally, seven methods (RDP, Chimaera, Boot-Scan, 3Seq, GENECONV, MaxChi, and SiScan) and three tree algorithms (Neighbor-joining, Max likelihood, and Bayesian) were employed for recombination analysis in RDP v4.101 (http://web.cbio.uct.ac.za/~darren/rdp.html). Recombination events characterized by more than three methods were considered reliable.

## Statistical analysis

The reading of sequencing data and the assembly of the MPXV genome sequence were done through the IPH nano sequencing analysis software. All quantitative data were summarized and plotted using Graph-Pad Prism software version 9.5.1 (https://www.graphpad-prism.cn/). The collection date of MPXV in IIb C.1 lineage was quantified and displayed in a four-digit format consisting of months and days. For the SNPs separated from the reference genome sequence of IIb B.1 (GenBank: ON563414.3) in the MPXV clusters of the IIb C.1 lineage, Shapiro–Wilk normality tests were used to verify whether continuous variables follow a normal distribution, and two-sided of independent sample T-test and One way ANOVA with Tukey's multiple comparisons test were used for comparison between two groups and multiple groups, respectively. $P > 0.05$ indicates no statistical significance, while $P < 0.05$ indicates a significant difference.

## Reporting summary

Further information on research design is available in the Nature Portfolio Reporting Summary linked to this article.

## Data availability

All analytical data are available within the article, Figures, and Supplementary Data. A total of 10 local MPXV whole genome sequences from outbreaks in Guangdong were included in this study. The assembled whole genome sequences have been deposited in the GenBase in the National Genomics Data Center, Beijing Institute of Genomics, Chinese Academy of Sciences/China National Center for Bioinformation, under accession number C_AA038923.1 to C_AA038932.1 that are publicly accessible at https://ngdc.cncb.ac.cn/genbase, and the raw sequencing data have been deposited in the Genome Sequence Archive (Genomics, Proteomics & Bioinformatics 2021) in National Genomics Data Center (Nucleic Acids Res 2022), China National Center for Bioinformation/Beijing Institute of Genomics, Chinese Academy of Sciences (GSA: CRA012147) that are publicly accessible at https://ngdc.cncb.ac.cn/gsa. The detailed information has been summarized and shown in Supplementary Data 1-Sheet 1.

## Code availability

All data analysis in this study was conducted using open-source software and websites presented in the Methods section. It does not involve the use of custom code or mathematical algorithms.

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

## Acknowledgements

We thank Juan Su, Qiqi Tan, and Changyun Sun from the Center for Disease Control and Prevention of Guangdong Province for their contributions in epidemiological investigations and sample collection. Professional english language editing was conducted by Editage (www.editage.cn). This work was supported by the National Natural Science Foundation of China (No. 82371846 [W.Z.] and No. 82173191 [B.L.]), Guangdong Science and Technology Program (No. 2021B1212030007 [B.L.]), Guangdong Science and Technology Program key projects (No. 2021B1212030014 [W.Z.]), Basic Research Project of Key Laboratory of Guangzhou (No. 202102100001 [W.Z.]), Yangjiang Science and Technology Program key projects (No. 2019010 [W.Z.]) and "Group-type" Special Supporting Project for Educational Talents in Universities (No. 4SG22264G [W.Z.]). The funders had no role in study design, data collection, and analysis, the decision to publish or preparation of the manuscript.

## Author contributions

J.Y., X.Z., J.J.L., C.S., W.Z., and B.L. contributed to research design and experimental planning. L.F., J.H., B.D., B.P., and J.T.L contributed to the organization of epidemiological data, sample collection, and sample processing. J.H., J.J.L., S.H., and M.Z. contributed to the sequencing and splicing of MPXV sequences. L.X., X.X., Y.L., and L.W. participated in bioinformatics analysis related to phylogenetic and molecular evolution. J.Y., X.Z., B.Z., C.S., W.Z., and B.L. were the major contributor to manuscript writing and language polishing. All authors have reviewed the content of the manuscript, agreed to be personally accountable for their own contributions and agreed to submit the final version.

## Competing interests

The authors declare no competing interests.
