## [Peer Review File · Nature Communications]

Phylogeny and molecular evolution of the first local monkeypox virus cluster in Guangdong Province, ChinaREVIEWER COMMENTS

Reviewer #1 (Remarks to the Author):

This is an important study that documents noteworthy findings on the evolution of monkeypox virus during outbreaks in China. The report is of excellent quality and the following are minor points to be addressed by the authors.

Major comment.

Discussion. is it possible to add a further statement as to the potential public health implications of the (APOBEC) mutations and recombinant strains found. Do these findings suggest the virus is adapting to greater ease of transmission, faster evolution, or other relevant possibilities? or is it too early to say what these findings may imply?

Minor comments

title. suggest removing the two capital M letters in the title as they are not necessary.

line 32. spell out APOBEC on first use - and provide definition in text or footnote.

line 38. historically endemic to East, Central and West Africa (add East - cf South Sudan Unity State outbreak of 2005)

line 46. suggest replacing 'typically' with 'often' as many countries are reporting local transmission during the global outbreak.

Some terminology adjustments are suggested to reduce stigmatization of persons at risk.

line 48. Replace 'infected individuals' with persons with mpox (or MPX) as a less stigmatizing term.

line 51. replace 'claimed' with 'reported' which is more neutral (and delete 'to')

line 53. delete 'importation'

line 70. replace 'caused by' with 'linked to'

line 79. meaning is unclear - 'local outbreak resulting from an importation event to the province from Beijing' would be clearer if this is the meaning.

Figure 1. change the legend label of hMpxV to MPXV for consistency with the rest of the paper and accepted practice.

Figure 1 C legend. spelling: unknown

line 157. replace 'hiddenly' with 'silently' or 'transmitted without detection'?

line 176. were these personal data also removed/delinked prior to analysis?

lines 215-218. no need to capitalize M in monkeypox virus (unless name is italicized)

Supplementary tables reviewed. no additional comment

Terminology. In 2023, MPX remains an accepted abbreviation for monkeypox and mpox is an alternative name for the disease. The choice lies with the authors. The name of the monkeypox virus (MPXV) remains unchanged and is correctly used in this paper.

References. OK.

Optional: consider including Ulaeto D et al. New nomenclature for mpox (monkeypox) and monkeypox virus clades. February 2023. The Lancet Infectious Diseases.23(3). DOI: 10.1016/S1473-3099(23)00055-5

Reviewer #2 (Remarks to the Author):

Dear authors,

First and foremost, I would like to congratulate the entire team for the exceptional work accomplished! Zoonotic diseases play a pivotal role in the realm of public health. Despite the reduction in its transmission rate, Mpox still harbors a multitude of unanswered questions, particularly considering its historical status as an endemic disease in Africa.

Regarding the manuscript, I believe it satisfactorily meets the criteria required for publication as a regional innovation, presenting a solid methodology and highly relevant outcomes. However, I would like to offer a few minor suggestions for adjustments below, aimed at enhancing the clarity and fortifying the article's foundation even further.

1. According to the WHO's recommendation, replace the term "monkeypox" in the name of the disease with "mpox" throughout the text.
2. Phylogenetic analysis: When compiling the dataset that originated Figure 1, it is necessary to provide the date on which this data was obtained from the platform.
3. The node reliability (bootstrap) was not included in the figure 1 and this is a very important data.
4. The colors in the legend are too similar, making it difficult to distinguish the elements in the figure 1. If adjusting the colors in a more intuitive manner isn't feasible, an alternative could be grouping similar underlines (for example, IIb A, A1, and A2, as well as A2.1, could be simplified to IIb. A). It's important to consider a solution that enhances the visualization and interpretation of the presented data.
5. In Supplementary Table 1 with the sequencing data, there is no information regarding the average depth and/or minimum depth (cut-off point) used in assembling the genomes to obtain the presented coverages.
6. Have you have constructed a tree containing complete genomes without masking non-coding and repetitive regions. Is there any difference among the sequences in these masked areas? If there is, even if we believe the impact on the organism to be negligible, it cannot be regarded as an absolute certainty. Preserving the complete genome would remain relevant, even if it entails computational costs.

Reviewer #3 (Remarks to the Author):

The study of Yu. et. al. shows an interesting insight into the mpox outbreak 2023. Several new outbreaks have been reported for summer 2023, including Europe, USA, Latin America, China and Japan. Although some of the virus genomes were uploaded, not much has been reported about the global outbreak of 2023 and the characteristics of the virus genomes. Furthermore, with only 114 reported cases in China, and few submitted virus genomes, the study includes some unique information regarding mpox in China.

Yu. et. al. report an outbreak with the same virus cluster in different regions of China, Japan (and recently also Portugal, not reported), indicating a connection between all events including hidden transmission. The cluster from Guangzhou, Shenzhen, and recently Portugal and Yunnan share some unique mutations and have the closest phylogenetic similarity to a strain from Japan, collected in April 2023.

General: The study shares interesting and important information about the local Guangzhou outbreak 2023. I think the importance of the study could be increased by:

- Including some recent information on the 2023 outbreak. Some strains have been added to the database last week, including strains from Portugal and Yunnan, some of them very similar to the Guangzhou strains, all lineage C.1 (new). Furthermore, the novel lineage B.1.20 has been sequenced in the USA. This enables a new perspective on the findings, as the same cluster was found in Portugal and China during the same time period.
- Including a broader perspective on the 2023 outbreak and genomes. Compared to the lineages of 2022, is the number of mutations in the 2023 genomes high or expected? Were the specific mutations of lineage C1 and the local strains also identified in the 2022 genomes? What is the difference of the 2023 outbreak genomes to 2022?

Methods: I was missing some information to evaluate the results. For example, some more info on sample preparation, CT values, more information on the sequencing and bioinformatics analysis. Was the sequencing approach shotgun, or amplicon.

Language: Although not a native speaker myself, I think that the language and syntax of the manuscript would need reviewing.

Ethics: It is stated that the study complies with ethical regulations and informed consent was given. However, I think that studies involving patients need an approval by an ethics committee. Also, ethics information in the reporting summary is given as "not applicable" (research involving human participants). I am also concerned about some of the authors statements like "cooperation level of patients (...) was suboptimal" or "the patients claim they did not know each other". This suggests to the reader that the authors think otherwise and blame the patients for not being cooperative and honest. This can be rephrased neutrally with "no information was available (..)" and "no connection between the patients was reported" without judging the patient. Concerning the latter citation ("the patients claim they did not know each other"), how was this information gained? I think this should also be explained or rephrased to avoid misunderstanding.

Specific:

- The authors use MPX as the term for the outbreak. I think that mpox should be used, as suggested by the WHO in 2022
- Line 26, "phylogenetic and molecular evolutionary", needs rephrasing
- Line 27, rapidly may give the impression of superficiality
- Line 29: "The (..) cluster showed a further evolutionary distance than that of Japan": Looking at all genomes from Japan from ~April 2023 I don't think this is true. Or is this in regard the reference used? Then rephrasing necessary for clarification.
- Line 29-31: whole sentence needs rephrasing, separated (by?)
- Line 30: This gives the impression that the whole cluster has 39 SNPS to the reference, which would be very high. Here, shared mutations could be given.
- Line 38: WHO also suggests not to use the term endemic, rather "previously affected" and "newly affected countries"
- Line 42: I think this is outdated, for 2023 website states "reported cases are low, frequency of reporting of cases has decreased substantially"
- Line 44 and 57: the first local outbreak in China?
- Line 65: The cluster has recently been updated to be labeled with C1. I think this should be updated, as this underlines the uniqueness of the lineages within all other B1 lineages
- Line 68-70: It is not clear to me if cases have been imported from Japan to Beijing or the other way around. Can maybe be rephrased to clarify
- Line 71: could you give the date for the Beijing cases?
- Line 75-76, see above, distance to the reference?
- Line 84: There are some strains from Yunnan and Portugal now in the database, highly similar to yours, might be worth to update
- Line 96, see above, gives the impression that rate of mutation is very high
- Line 97: the terms for mutations "same" and "partial" are very uncommon, maybe use shared, partially shared or similar terms?
- Figure 2A: please include that you used the coordinates of NC_063383 for the reference ON563414.3
- Figure 2A and 2D: Most of the MPXV 2022 mutations were APOBEC, here a higher number of the private or partially shared mutations are non-APOBEC. It would be interesting to show the

confidence for the bases at these positions, for example the coverage and number of reads, to support the finding.

- Figure 2C: Frequency rather in %?
- Figure 2F: needs more explanation in the text, see below
- Line 124-125: The meaning of the sentence is unclear to me, which mutations?
- Line 141-152: The whole passage needs more context and explanation: What kind of recombination do the strains have? Where? And why is that important? More description and/or graphic. If all B.1 strains have the recombination, isn't it more interesting that only one of your strains has it?
- Line 152: I would not say that morbidity is increasing, only cases in and relation to beginning of 2023
- Line 155: Importation events in China
- Line 160: it is a bit speculative that the mutations have significant implications, maybe relativize a bit
- Line 163: mpox outbreak
- Methods: It would be interesting to have more information about the samples taken. Why was Serum taken and not swabs? Are there any CT values? Also some info on sample preparation (Kits used etc)
- Methods: Also some more information on sequencing and data analysis could be given, for example how many reads/viral reads were sequenced, and how the genomes were assembled

Reviewer #4 (Remarks to the Author):

The authors Yu et al. describe the genomic investigation of Mpox sequences from the Guangdong province of China. The authors describe the phylogenetic clustering of their sequences with other sequences from China and with a large outbreak in Japan. This report presents valuable genomic epidemiological study of a Mpox outbreaks. However, the study and results description can be improved to clarify the context and give confidence to the results. Below are more specific comments. I would encourage this report to be published after careful revision of the following points.

1. Write mpox instead of monkeypox. See references below:
 - a. Happi, Christian, et al. "Urgent need for a non-discriminatory and non-stigmatizing nomenclature for monkeypox virus." *PLoS biology* 20.8 (2022): e3001769.
 - b. Ulaeto, David, et al. "New nomenclature for mpox (monkeypox) and monkeypox virus clades." *The Lancet Infectious Diseases* 23.3 (2023): 273-275.
2. Line 48 – "However, in this study, the cooperation level of patients during the epidemiological investigation was suboptimal." – Not clear what this means
3. Generally, it would help to see an epidemiological curve of the number of recorded cases through time in China and in the Guangdong province (potential linked to the timing of sequence sampling dates) to more clearly understand the proportion of cases sequenced and the epidemiological context that the genomes belong to.
4. Sequences need to be made public in Genbank before publication
5. Line 50 – "the patients had no overseas travel history;" but Table S1 shows travel history of one patient to Hong Kong
6. Figure 1: when describing "evolutionary distance was further than Japan sequences" in main text and in the figure legend, please specify if you mean "from the root".
7. It would be important to check again if the Beijing sequences are now available to update the results. Seems like an important missing piece of the puzzle.
8. I would interpret the phylogenetics results as an introduction from Japan via Zhejiang province and then into Guangdong. Having sequenced the Zhejiang sequence at similar dates as the Guangdong province does not eliminate this option and to me, the tree really show that the introduction to Guangdong came rather from Zhejiang. Again this is where bootstrap values or Bayesian inference could help.
9. It would be good to add a root-to-tip substitution plot for the Japan, Zhejiang, Guangdong sequences to characterize the molecular clock evolution

10. Methods:

- a. Not enough details on the sampling – for e.g. two of the sequenced specimens came from 3rd passage in Vero E6 culture – why? This process could add spurious mutations specific to cell culture so the authors should explain why this was done and if the sequences were analyzed for any additional mutations
- b. Not enough details on genome sequencing process, especially if this sequencing was done in-house
- c. Not enough details on the assembly process – please reference “IPH-nano”. Please include assembly parameters like depth of coverage etc
- d. Phylogenetics
 - i. ML tree – were bootstraps used? There is no indication of such on the tree. This could be important to interpret confidence in clusters.
- e. Molecular evolution analysis – no need to say “rapidly” extracted when describing the use of Nextclade

Dear Reviewers,

Thank you for your helpful insights into our manuscript. We have studied all the comments carefully and have made revision by marking the corresponding changes in red font in the revised manuscript for your ease of review. The following are the specific revision details:

REVIEWER COMMENTS

Reviewer #1 (Remarks to the Author):

This is an important study that documents noteworthy findings on the evolution of monkeypox virus during outbreaks in China. The report is of excellent quality and the following are minor points to be addressed by the authors.

Major comment.

1. Discussion. is it possible to add a further statement as to the potential public health implications of the (APOBEC) mutations and recombinant strains found. Do these findings suggest the virus is adapting to greater ease of transmission, faster evolution, or other relevant possibilities? or is it too early to say what these findings may imply?

Reply: Thank you for your suggestion. To strengthen the discussion regarding the molecular evolution analysis results, we have added further explanation in the corresponding sections describing microevolution and recombination analysis (lines 146-152, 184-191, 204-210, and 213-216) and added a paragraph to further illustrate the significance of the discovered (APOBEC) mutations and the recombinant event in lines 220-232.

Minor comments

1. title. suggest removing the two capital M letters in the title as they are not necessary.

Reply: Thank you for your suggestion. We have made modifications to the title in line 1.

2. line 32. spell out APOBEC on first use - and provide definition in text or footnote.

Reply: Thank you for your reminder. We have spelled out the full name of APOBEC on the abstract on line 35 and added its definition and functional description in the text at lines 139-142.

3. line 38. historically endemic to East, Central and West Africa (add East - cf South Sudan Unity State outbreak of 2005)

Reply: Based on the information you provided, we have added “East” in line 42.

4. line 46. suggest replacing 'typically' with 'often' as many countries are reporting local transmission during the global outbreak.

Reply: We have replaced this word in line 52.

Some terminology adjustments are suggested to reduce stigmatization of persons at risk.

1. line 48. Replace 'infected individuals' with persons with mpox (or MPX) as a less stigmatizing term.

Reply: Thank you for your suggestion. We have replaced ‘infected individuals’ with ‘individuals infected with MPX’ in line 53-54.

2. line 51. replace 'claimed' with 'reported' which is more neutral (and delete 'to')

Reply: We have revised this sentence based on your suggestion in line 58.

3. line 53. delete 'importation'

Reply: We have deleted this word based on your suggestion in line 60.

4. line 70. replace 'caused by' with 'linked to'

Reply: We have replaced this word based on your suggestion in line 98.

5. line 79. meaning is unclear - 'local outbreak resulting from an importation event to the province from Beijing' would be clearer if this is the meaning.

Reply: We apologize for the inconvenience and have changed this sentence to "However, whether this local outbreak in Guangdong resulted from the importation event that occurred in Beijing or multiple transmission chains caused by different importation events from Japan, remains uncertain" to clearly express the meaning of this sentence in lines 111-113.

6. Figure 1. change the legend label of hMpxV to MPXV for consistency with the rest of the paper and accepted practice.

Reply: We have changed it to MPXV in Figure 1e.

7. Figure 1 C legend. spelling: unknown

Reply: We have added a description in the legend of Figure 2d (previously displayed as Figure 1 C) in line 136-137, explaining that 'Unknown' means the GenBank function of this protein was annotated as "unknown" in the reference MPXV (GenBank access no. NC_063383.1).

8. line 157. replace 'hiddenly' with 'silently' or 'transmitted without detection'?

Reply: We have replaced "hiddenly transmission" with "silent transmission" throughout the revised manuscript, based on your suggestion.

9. line 176. were these personal data also removed/delinked prior to analysis?

Reply: We would like to confirm that the importance of protecting patient information was considered and personal information was deleted by the Guangdong Provincial Center for Disease Control and Prevention. The entire analysis process of this study was conducted anonymously. We have added a description in the manuscript to emphasize this issue in lines 257-259.

10. lines 215-218. no need to capitalize M in monkeypox virus (unless name is italicized)

Reply: We thank you for your suggestion and have made corresponding modifications in lines 317-319 and Supplementary Data 1.

11. Terminology. In 2023, MPX remains an accepted abbreviation for monkeypox and mpox is an alternative name for the disease. The choice lies with the authors. The name of the monkeypox virus (MPXV) remains unchanged and is correctly used in this paper.

Reply: We apologize that the terminology has not been updated. In the revised version, we have replaced monkeypox with mpox (MPX) to represent the disease name, while retaining monkeypox virus (MPXV) to represent the virus name.

12. Optional: consider including Ulaeto D et al. New nomenclature for mpox (monkeypox) and monkeypox virus clades. February 2023. The Lancet Infectious Diseases.23(3). DOI: 10.1016/S1473-3099(23)00055-5

Reply: Thank you for your suggestion. We have added it as reference 3 to the new version.

Reviewer #2 (Remarks to the Author):

Dear authors,

First and foremost, I would like to congratulate the entire team for the exceptional work accomplished! Zoonotic diseases play a pivotal role in the realm of public health. Despite the reduction in its transmission rate, Mpox still harbors a multitude of unanswered questions, particularly considering its historical status as an endemic disease in Africa.

Regarding the manuscript, I believe it satisfactorily meets the criteria required for publication as a regional innovation, presenting a solid methodology and highly relevant outcomes. However, I would like to offer a few minor suggestions for adjustments below, aimed at enhancing the clarity and fortifying the article's foundation even further.

1. According to the WHO's recommendation, replace the term "monkeypox" in the name of the disease with "mpox" throughout the text.

Reply: We apologize that the terminology has not been updated. In the revised version, we have replaced monkeypox with mpox (MPX) to represent the disease name, while retaining monkeypox

virus (MPXV) to represent the virus name.

2. Phylogenetic analysis: When compiling the dataset that originated Figure 1, it is necessary to provide the date on which this data was obtained from the platform.

Reply: Thank you for your reminder. We have now added the date for obtaining the dataset of MPXV sequences from the GISAID database in the method section (line 277) and Figure 1 (line 84).

3. The node reliability (bootstrap) was not included in the figure 1 and this is a very important data.

Reply: Thank you for your suggestion. We have described the method used for calculating node support rate in the method section (lines 289-290) and showed it in the corresponding section of Figure 1e.

4. The colors in the legend are too similar, making it difficult to distinguish the elements in the figure 1. If adjusting the colors in a more intuitive manner isn't feasible, an alternative could be grouping similar underlines (for example, Iib A, A1, and A2, as well as A2.1, could be simplified to Iib. A). It's important to consider a solution that enhances the visualization and interpretation of the presented data.

Reply: We apologize for the poor visuals in Figure 1. We have adjusted the color allocation and interpretation process of the phylogenetic tree in the revised Figure 1, hoping to demonstrate the phylogenetic analysis process of the Iib C.1 lineage that we focused on, in our study, by expanding layer-by-layer.

5. In Supplementary Table 1 with the sequencing data, there is no information regarding the average depth and/or minimum depth (cut-off point) used in assembling the genomes to obtain the presented coverages.

Reply: We apologize for missing this information. We have added more detailed genome assembly data in the revised Supplementary Table 1, including coverage, total number of reads, read length, and mean depth of coverage.

6. Have you have constructed a tree containing complete genomes without masking non-coding and repetitive regions. Is there any difference among the sequences in these masked areas? If there is, even if we believe the impact on the organism to be negligible, it cannot be regarded as an absolute certainty. Preserving the complete genome would remain relevant, even if it entails computational costs.

Reply: Thank you for your question. We chose to mask non-coding and repetitive regions not to reduce computational complexity but to maintain consistency with the process of constructing phylogenetic trees in GISAID and NextClade. Although the phylogenetic analysis process of NextClade (<https://nextstrain.org/monkeypox/hmpxv1?dmin=2021-07-03>) does not explain why these regions were masked, we believe it may be related to the tendency of the end region of the monkeypox virus genome to be error-prone during sequencing, resulting in an excess of inaccurate mutations, as mentioned in Joana Isidro et al.'s study (Phylogenomic characterization and signs of microevolution in the 2022 multi-country outbreak of monkeypox virus. *Nature Medicine* vol. 28,8 (2022): 1569–1572). Monkeypox viruses with excessive mutations in their terminal genomes require further masking before conducting phylogenetic analysis. Therefore, to avoid interference by inaccurate mutations at non-coding and repetitive regions of some MPXV genome sequences obtained from the GISAID database that may lead to bias in phylogenetic analysis, we masked these regions before constructing the phylogenetic tree.

Reviewer #3 (Remarks to the Author):

The study of Yu. et. al. shows an interesting insight into the mpox outbreak 2023. Several new outbreaks have been reported for summer 2023, including Europe, USA, Latin America, China and Japan. Although some of the virus genomes were uploaded, not much has been reported about the global outbreak of 2023 and the characteristics of the virus genomes. Furthermore, with only 114 reported cases in China, and few submitted virus genomes, the study includes some unique information regarding mpox in China.

Yu. et. al. report an outbreak with the same virus cluster in different regions of China, Japan (and recently also Portugal, not reported), indicating a connection between all events including hidden transmission. The cluster from Guangzhou, Shenzhen, and recently Portugal and Yunnan share some unique mutations

and have the closest phylogenetic similarity to a strain from Japan, collected in April 2023.

General: The study shares interesting and important information about the local Guangzhou outbreak 2023. I think the importance of the study could be increased by:

1. Including some recent information on the 2023 outbreak. Some strains have been added to the database last week, including strains from Portugal and Yunnan, some of them very similar to the Guangzhou strains, all lineage C.1 (new). Furthermore, the novel lineage B.1.20 has been sequenced in the USA. This enables a new perspective on the findings, as the same cluster was found in Portugal and China during the same time period.

Reply: To incorporate your suggestion and gain a more comprehensive understanding of the relationship between the recently updated MPXV sequences and that of Guangdong, we focused on the outbreak of the lineage IIb C.1 in Guangdong Province and considered the latest 62 MPXV sequences from South Korea, Japan, Portugal, and Yunnan and Shenzhen of China to construct a new phylogenetic tree, shown in revised Figure 1.

2. Including a broader perspective on the 2023 outbreak and genomes. Compared to the lineages of 2022, is the number of mutations in the 2023 genomes high or expected? Were the specific mutations of lineage C1 and the local strains also identified in the 2022 genomes? What is the difference of the 2023 outbreak genomes to 2022?

Reply: After discussing with all authors, based on the updated phylogenetic tree, we described the phylogenetic characteristics with temporal continuity of MPXV in 2022 and 2023 and further analyzed the genomic evolution characteristics of MPXV in different periods, to better elucidate the propagation and evolutionary process of lineage IIb C.1. The detailed results and discussions are presented in lines 73–81, 157-168 and 182-200.

3. Methods: I was missing some information to evaluate the results. For example, some more info on sample preparation, CT values, more information on the sequencing and bioinformatics analysis. Was the sequencing approach shotgun, or amplicon.

Reply: We apologize for missing this information in the methods section. Based on your suggestion, we have added a more complete description of the process of sample preparation, sequencing, and genome assembly in the methods section (lines 262-268). Furthermore, the epidemiological background of the samples and detailed data on genome assembly are compiled in Table S1.

4. Language: Although not a native speaker myself, I think that the language and syntax of the manuscript would need reviewing.

Reply: Thank you for your suggestion. To further review the language and syntax of the manuscript, we submitted the revised manuscript after making revisions based on the comments of all reviewers to Editage (www.editage.cn), before submitting the final version on the submission system of *Nature Communications*.

5. Ethics: It is stated that the study complies with ethical regulations and informed consent was given. However, I think that studies involving patients need an approval by an ethics committee. Also, ethics information in the reporting summary is given as “not applicable” (research involving human participants). I am also concerned about some of the authors statements like “cooperation level of patients (...) was suboptimal” or “the patients claim they did not know each other”. This suggests to the reader that the authors think otherwise and blame the patients for not being cooperative and honest. This can be rephrased neutrally with “no information was available (..)” and “no connection between the patients was reported” without judging the patient. Concerning the latter citation (“the patients claim they did not know each other”), how was this information gained? I think this should also be explained or rephrased to avoid misunderstanding.

Reply: We are aware of the lack of information on ethics approval, therefore, we have now separated the 'Ethics Statement' in the method section and added a more complete description of the ethical approval and anonymous analysis process in lines 252-259. To rectify the ambiguity caused by sentences describing epidemiological survey results, we have also made corresponding corrections (lines 54-58).

Specific:

1. The authors use MPX as the term for the outbreak. I think that mpox should be used, as suggested by the WHO in 2022

Reply: We apologize that the terminology has not been updated. In the revised version, we have replaced monkeypox with mpox (MPX) to represent the disease name, while retaining monkeypox virus (MPXV) to represent the virus name.

2. Line 26, “phylogenetic and molecular evolutionary”, needs rephrasing; Line 27, rapidly may give the impression of superficiality; Line 29: “The (..) cluster showed a further evolutionary distance than that of Japan”: Looking at all genomes from Japan from ~April 2023 I don’t think this is true. Or is this in regard the reference used? Then rephrasing necessary for clarification; Line 29-31: whole sentence needs rephrasing, separated (by?); Line 30: This gives the impression that the whole cluster has 39 SNPS to the reference, which would be very high. Here, shared mutations could be given.

Reply: We have revised the sentences and words in the abstract based on your detailed suggestions in lines 25-36.

7. Line 38: WHO also suggests not to use the term endemic, rather “previously affected” and “newly affected countries”

Reply: We have replaced this word in line 42-43, based on your suggestion.

8. Line 42: I think this is outdated, for 2023 website states “reported cases are low, frequency of reporting of cases has decreased substantially“

Reply: We apologize for the incorrect description. We have re-edited this paragraph based on the latest report from the World Health Organization (https://worldhealthorg.shinyapps.io/mpx_global/) in lines 46-49.

9. Line 44 and 57: the first local outbreak in China?

Reply: We wanted to state that the first local MPX outbreak recorded in Guangdong Province was

in June 2023; therefore, we have made a correction in line 50.

10. Line 65: The cluster has recently been updated to be labeled with C1. I think this should be updated, as this underlines the uniqueness of the lineages within all other B1 lineages

Reply: We have updated the latest lineage in line 92.

11. Line 68-70: It is not clear to me if cases have been imported from Japan to Beijing or the other way around. Can maybe be rephrased to clarify

Reply: We have rephrased this sentence (lines 95-104).

12. Line 71: could you give the date for the Beijing cases?

Reply: We have added the collection date for the two cases from Beijing in line 101.

13. Line 75-76, see above, distance to the reference?

Reply: We apologize for the error in this description. After further analysis of the phylogenetic tree, we found that the description of this sentence is incorrect. The branch length of the phylogenetic tree we constructed can only represent the degree of nucleotide differences between MPXV genomes and cannot reflect evolutionary distance. Therefore, we have deleted this sentence.

14. Line 84: There are some strains from Yunnan and Portugal now in the database, highly similar to yours, might be worth to update

Reply: Thank you for your suggestion. To gain a more comprehensive understanding of the relationship between the recently updated MPXV sequences and that of Guangdong, we focused on the outbreak of lineage IIb C.1 in Guangdong Province and added the latest 62 MPXV sequences from South Korea, Japan, Portugal, and Yunnan and Shenzhen of China to construct a new phylogenetic tree in the revised Figure 1.

15. Line 96, see above, gives the impression that rate of mutation is very high

Reply: Thank you for your suggestion. To avoid this issue, we have modified this sentence (line 121) and adjusted the further analysis process. First, we described that the 10 MPXV sequences of Guangdong diverged by a mean of 21 SNPs. Second, we analyzed the distribution characteristics of 39 distinct SNPs obtained from the Guangdong cluster in this study. Finally, we analyzed a unique mutation spectrum of 14 shared mutations with time continuity molecular evolution in South Korea, Japan, China, and Portugal, which share phylogenetic relationships.

16. Line 97: the terms for mutations “same” and “partial” are very uncommon, maybe use shared, partially shared or similar terms?

Reply: We have uniformly revised these terms to “shared mutations,” “partially shared mutations,” and “private mutations,” based on your suggestion.

17. Figure 2A: please include that you used the coordinates of NC_063383 for the reference ON563414.3

Reply: Thank you for your reminder. We have added this description to the legend of Figure 2a in line 132-133.

18. Figure 2A and 2D: Most of the MPXV 2022 mutations were APOBEC, here a higher number of the private or partially shared mutations are non-APOBEC. It would be interesting to show the confidence for the bases at these positions, for example the coverage and number of reads, to support the finding.

Reply: Thank you for your suggestion. We also noticed this phenomenon and found it in a previous analysis of the MPXV outbreak in 2022; however, this was not specifically observed and described. Therefore, we have added a description and explanation of this phenomenon in the results section in lines 146-152. In addition, based on your suggestion, we repeatedly verified the assembly process of the MPXV genome sequence in this study to ensure the accuracy of our sequence and uploaded the original sequencing data to a public database for readers to access.

19. Figure 2C: Frequency rather in %?

Reply: We have corrected this mistake in Figure 2d (previously displayed as Figure 2C).

20. Figure 2F: needs more explanation in the text, see below

Reply: We have re-display the results of recombination analysis in Figure 3c-3e, and deeply described and discussed the results in lines 202-216 and 225-232, based on your comment.

21. Line 124-125: The meaning of the sentence is unclear to me, which mutations?

Reply: We apologize for the ambiguity in the meaning of the sentence and have revised this description in lines 149-152.

22. Line 141-152: The whole passage needs more context and explanation: What kind of recombination do the strains have? Where? And why is that important? More description and/or graphic. If all B.1 strains have the recombination, isn't it more interesting that only one of your strains has it?

Reply: Thank you for your suggestion. As mentioned above, we have re-display the results of recombination analysis in Figure 3c-3e, and deeply described and discussed the results in lines 202-216 and 225-232, based on your comment.

23. Line 152: I would not say that morbidity is increasing, only cases in and relation to beginning of 2023; Line 155: Importation events in China; Line 160: it is a bit speculative that the mutations have significant implications, maybe relativize a bit; Line 163: mpox outbreak

Reply: We have revised these sentences in lines 219, 236, 240-242 and 244.

24. Methods: It would be interesting to have more information about the samples taken. Why was Serum taken and not swabs? Are there any CT values? Also some info on sample preparation (Kits used etc);

Methods: Also some more information on sequencing and data analysis could be given, for example how many reads/viral reads were sequenced, and how the genomes were assembled.

Reply: We apologize that the sample type was written as serum, as this was an error caused during data collection and summary. All the samples first used for sequencing were herpes collection fluid and, because of insufficient sequencing depth and coverage, the complete MPXV genome sequence could not be assembled using the herpes collection fluid for patients M23011 and M23008. Therefore, these two MPXV sequences were sequenced and assembled from the first-generation virus isolated from herpes collection fluid. We have updated this part in Table S1.

In addition, we apologize for the missing information on methods. We have added a more complete description of the process of sample preparation, sequencing, and genome assembly in the methods section in lines 262-268. Furthermore, the epidemiological background of the samples and detailed data on genome assembly are compiled in Table S1.

Reviewer #4 (Remarks to the Author):

The authors Yu et al. describe the genomic investigation of Mpox sequences from the Guangdong province of China. The authors describe the phylogenetic clustering of their sequences with other sequences from China and with a large outbreak in Japan. This report presents valuable genomic epidemiological study of a Mpox outbreaks. However, the study and results description can be improved to clarify the context and give confidence to the results. Below are more specific comments. I would encourage this report to be published after careful revision of the following points.

1. Write mpox instead of monkeypox. See references below:

a. Happi, Christian, et al. "Urgent need for a non-discriminatory and non-stigmatizing nomenclature for monkeypox virus." *PLoS biology* 20.8 (2022): e3001769.

b. Ulaeto, David, et al. "New nomenclature for mpox (monkeypox) and monkeypox virus clades." *The Lancet Infectious Diseases* 23.3 (2023): 273-275.

Reply: We apologize that the terminology has not been updated. In the revised version, we have replaced monkeypox with mpox (MPX) to represent the disease name, while retaining monkeypox virus (MPXV) to represent the virus name.

2. Line 48 – “However, in this study, the cooperation level of patients during the epidemiological investigation was suboptimal.” – Not clear what this means

Reply: We apologize for the ambiguity in sentences that describe epidemiological survey results. We made corresponding corrections in lines 54-55.

3. Generally, it would help to see an epidemiological curve of the number of recorded cases through time in China and in the Guangdong province (potential linked to the timing of sequence sampling dates) to more clearly understand the proportion of cases sequenced and the epidemiological context that the genomes belong to.

Reply: Thank you for your suggestion. We apologize for not having the available data to plot an epidemiological curve of the number of recorded cases over time in China and in the Guangdong province. However, the 10 MPXV sequences analyzed in our study were from the first 10 confirmed MPX patients reported in Guangdong Province. We believe that the phylogenetic and molecular evolutionary characteristics of these sequences from an early outbreak will help provide insights about the source and the silent transmission of the first reported local MPX from Guangdong.

4. Sequences need to be made public in Genbank before publication

Reply: Thank you for pointing out this issue. We also understand the importance of publicly available sequences. Therefore, during the submission, we uploaded the assembled 10 MPXV genome sequences and their raw sequencing data to the public database and updated the accession numbers in Table S1. A data availability statement has been added to the revised manuscript (lines 345-354).

5. Line 50 – “the patients had no overseas travel history;” but Table S1 shows travel history of one patient to Hong Kong

Reply: We have revised this sentence in lines 56-57.

6. Figure 1: when describing “evolutionary distance was further than Japan sequences” in main text and in the figure legend, please specify if you mean “from the root”.

Reply: We apologize for the error in this description. After further analysis of the phylogenetic tree, we found that the description of this sentence is incorrect. The branch length of the phylogenetic tree we constructed can only represent the degree of nucleotide differences between MPXV genomes and cannot be used to reflect evolutionary distance. Therefore, we have deleted this sentence.

7. It would be important to check again if the Beijing sequences are now available to update the results. Seems like an important missing piece of the puzzle.

Reply: We strongly agree with your suggestion. However, Daitao Zhang et al. did not provide corresponding public database accession numbers in their study for the two Beijing MPX cases, and we have searched known public databases such as GISAID, Genbank, GenBase, etc. but could not find information on MPXV from Beijing. Therefore, we discussed the impact of the absence of MPXV in Beijing in the revised manuscript (lines 100-113 and 187-192), hoping to provide guidance for future research.

8. I would interpret the phylogenetics results as an introduction from Japan via Zhejiang province and then into Guangdong. Having sequenced the Zhejiang sequence at similar dates as the Guangdong province does not eliminate this option and to me, the tree really show that the introduction to Guangdong came rather from Zhejiang. Again this is where bootstrap values or Bayesian inference could help.

Reply: Thank you for raising the key question. After further analysis, we found that, in the absence of the two first reported cases of Beijing MPXV in Chinese Mainland, the phylogenic and molecular

evolution characteristics indicated that the Zhejiang strain was reported earlier and was a more likely source of the outbreak of Guangdong MPX. Therefore, we have provided a more detailed description and discussion of this issue in the revised manuscript in lines 105-110 and 187-195.

9. It would be good to add a root-to-tip substitution plot for the Japan, Zhejiang, Guangdong sequences to characterize the molecular clock evolution

Reply: Based on your suggestion, we have divided the MPXV of the complete lineage IIb C.1 into virus clusters with time continuity based on phylogenetic location and conducted a root-to-tip molecular microevolution analysis to elucidate the unique evolutionary mutation spectrum of the IIb C.1 lineage. The detailed results and discussions were added in lines 157-168 and 182-200.

10. Methods:

a. Not enough details on the sampling – for e.g. two of the sequenced specimens came from 3rd passage in Vero E6 culture – why? This process could add spurious mutations specific to cell culture so the authors should explain why this was done and if the sequences were analyzed for any additional mutations

Reply: We apologize for missing the information on the sample source. To address this issue, the sample source information has been updated in Table S1. We explained the reason in the table annotations for using isolated viruses instead of Herpes collection fluid for sequencing analyses of M23011 and M23008 patients.

b. Not enough details on genome sequencing process, especially if this sequencing was done in-house;

c. Not enough details on the assembly process – please reference “IPH-nano”. Please include assembly parameters like depth of coverage etc; d. Phylogenetics

Reply: We apologize for the missing information on methods. We have added a more complete description of the process of sample preparation, sequencing, and genome assembly in the methods section (lines 262-268). Furthermore, the epidemiological background of the samples and detailed data on genome assembly are compiled in Table S1.

i. ML tree – were bootstraps used? There is no indication of such on the tree. This could be important to interpret confidence in clusters.

Reply: Thank you for your suggestion. We have described the method used for calculating node support rate in the method section (lines 289-290) and showed it in the corresponding section of Figure 1e.

e. Molecular evolution analysis – no need to say “rapidly” extracted when describing the use of Nextclade

Reply: We have removed this word from the corresponding sentence (line 293).

REVIEWERS' COMMENTS

Reviewer #3 (Remarks to the Author):

Dear Authors,

thank you for your great work and regarding all comments made. I have no further objections!

Kind regards